# Fermentation by Multiple Bacterial Strains Improves the Production of Bioactive Compounds and Antioxidant Activity of Goji Juice

**DOI:** 10.3390/molecules24193519

**Published:** 2019-09-28

**Authors:** Yuxuan Liu, Huan Cheng, Huiyan Liu, Ruoshuang Ma, Jiangtao Ma, Haitian Fang

**Affiliations:** 1College of Agriculture, Ningxia University, Yinchuan 750021, China; lyxlearn@163.com (Y.L.); f3112@sina.com (H.L.); majiangtao0214@163.com (J.M.); 2Ningxia Key Laboratory for Food Microbial-Applications Technology and Safety Control, Ningxia University, Yinchuan 750021, China; 3College of Biosystems Engineering and Food Science, National-Local Joint Engineering Laboratory of Intelligent Food Technology and Equipment, Zhejiang Key Laboratory for Agro-Food Processing, Zhejiang Engineering Laboratory of Food Technology and Equipment, Zhejiang University, Hangzhou 310058, China; huancheng@zju.edu.cn

**Keywords:** goji juice, fermentation, multiple bacterial strains, volatile compound, antioxidant capacity

## Abstract

Microorganisms can be used for enhancing flavors or metabolizing functional compounds. The fermented-food-derived bacterial strains comprising *Bacillus velezensis*, *Bacillus licheniformis*, and *Lactobacillus reuteri* mixed with *Lactobacillus rhamnosus* and *Lactobacillus plantarum* were used to ferment goji berry (*Lycium barbarum* L.) juice in this study. The fermentation abilities and antioxidant capacities of different mixtures of multiple strains in goji juice were compared. The results showed that the lactic acid contents increased 9.24–16.69 times from 25.30 ± 0.71 mg/100 mL in goji juice fermented using the SLV (*Lactobacillus rhamnosus*, *Lactobacillus reuteri*, and *Bacillus velezensis*), SZP (*Lactobacillus rhamnosus*, *Lactobacillus plantarum*, and *Bacillus licheniformis*), and SZVP (*Lactobacillus rhamnosus*, *Lactobacillus plantarum*, *Bacillus velezensis*, and *Bacillus licheniformis*) mixtures, and the protein contents increased 1.31–2.11 times from 39.23 ± 0.67 mg/100 mL. In addition, their contents of volatile compounds increased with positive effects on aroma in the fermented juices. Conversion of the free and bound forms of phenolic acids and flavonoids in juice was influenced by fermentation, and the antioxidant capacity improved significantly. Fermentation enhanced the contents of lactic acid, proteins, volatile compounds, and phenols. The antioxidant capacity was strongly correlated with the phenolic composition.

## 1. Introduction

The goji berry or wolfberry (*Lycium barbarum* L.) is an herb and a popular healthy food. Goji berries are rich in polysaccharides, carotenoids, phenolic acids, flavonoids, and betaine. Studies have showed various functional effects of goji berries, such as a hypoglycemic effect [1], antioxidant activity [2], and liver protection [3]. Thus, the goji berry is considered valuable in the food industry.

Foods fermented by microorganisms have unique flavors, and fermented foods represent a large segment of the food market at present. Several studies have determined the functions, microbial compositions, and nutritional contents of fermented foods [4,5,6]. Some of the metabolites produced by microorganisms can be released during fermentation to improve the flavor and nutritional value of foods [7]. Lactic acid bacteria and yeast are the most common microorganisms used for the fermentation of plant foods. In addition, some microorganisms isolated from fermented plant products were found to make important contributions to flavor and nutritional improvements [6,8].

Various fermented goji products including wine, vinegar, and yogurt have been investigated in recent studies [9,10,11]. Indeed, the fermentation of goji has become a key research area. To the best of our knowledge, the studies on fermented goji products focused mainly on the changes of the nutrients during the fermentation process. Not many results have been reported in the use of strains, and the differences in the fermented goji using different strains were not compared. The aim of this study was thus to compare the differences of the goji juices fermented by three mixed-strain groups. The properties of the fermented goji juices including chemical composition and volatile composition were evaluated. The change in phenolic compounds and antioxidant activity were also evaluated in fermented goji juice as compared with the raw juice. The results of this study provide novel insights into the production of fermented goji juice and the effects of different bacterial strains on the phenolic contents of goji juice.

## 2. Results and Discussion

### 2.1. Fermentation Characteristics of Goji Juice

As shown in Table 1, the concentration of reducing sugars was 5521.04 ± 4.58 mg/100 mL in the raw juice and it decreased by 11.53%, 16.61%, and 28.43% in the goji juice samples fermented with SLV, SZP, and SZVP, respectively, indicating that most of the saccharides present in the goji juice samples were consumed during fermentation. The saccharides in the fermented juice were utilized as carbon sources by the bacterial strains to generate lactic acid and other chemicals via a series of chemical reactions. The lactic acid concentrations were 25.30 ± 0.71 mg/100 mL in the raw juice, and 316.93 ± 7.31, 233.8 ± 10.97, and 422.14 ± 6.52 mg/100 mL in the goji juice samples fermented with SLV, SZP, and SZVP, respectively. The concentration of lactic acid was higher in the juice fermented with SZVP than the other samples. Compared with SZP, the addition of *Bacillus velezensis* greatly enhanced the capacity to produce lactic acid. The SLV group utilized reducing sugars at a lower rate than SZP, but the concentration of lactic acid produced after fermentation was 1.36 times that in SZP. Thus, the strains in SLV may have directly utilize the sucrose in goji juice to produce lactic acid. These observations agree with a previous study where cashew apple juice was fermented with *Lactobacillus plantarum*, *L. acidophilus,* and *L. casei* [12], and different sugars were directly utilized by different bacteria.

The concentrations of soluble proteins were 39.23 ± 0.67 mg/100 mL in the unfermented goji juice, and increased by 90.54%, 31.18%, and 110.91% in the goji juice samples fermented with SLV, SZP, and SZVP, respectively. Arte et al. [13] also obtained a similar result from fermentation of wheat bran with *L. plantarum* and yeast, where they found that fermentation increased the protein content of the fermented wheat bran by 20%. Some bacterial strains produced or were activated by a variety of enzymes, such as endogenous proteases during fermentation [14]. Proteases can break down macromolecular proteins and convert them into small protein molecules. It is possible that the soluble protein contents increased in the goji juice samples due to the action of proteases. The soluble protein contents were higher in the juices fermented with SZVP and SLV, which may be related to the presence of *B. velezensis* and *L. reuteri*. It was reported that *B. velezensis* can produce bacteriostatic substances comprising cyclic lipoprotein peptides during fermentation [15].

The number of live bacteria was slightly lower in the juice fermented with SLV compared with that in the juices fermented with SZP and SZVP, where the cell numbers in both exceeded 10^9^ CFU/mL. Zhang et al. [16] showed that some strains can produce endoglucanases during fermentation, which can degrade cellulose into cellooligosaccharides, and the growth of *L. plantarum* can be promoted by this process. *Lactobacillus plantarum* and *B. licheniformis* were present in the SZP and SZVP mixtures, but not in the SLV fermentation mixture. Therefore, it is proposed that *B. licheniformis* may produce a cellulase similar to endoglucanase to promote the growth of *L. plantarum*. The number of live bacteria was higher in the fermentation mixture containing *L. plantarum* and *B. licheniformis*.

The sensory quality scores were clearly improved after fermentation compared with the raw juice. Fermentation did not cause color change and unpleasant odor production in the goji juice, but the taste was improved significantly. There was no significant difference in the sensory quality of the goji juices fermented with SLV and SZVP, but their sensory quality scores were better than that for the juice fermented with SZP.

Overall, the juices fermented with SZVP and SLV groups had a greater capacity for acid production and higher soluble protein contents. The strains present in these mixtures could grow normally in the goji juice, and the sensory quality was better after fermentation. The SZVP and SLV mixtures had a greater capacity for fermentation than SZP. 

### 2.2. Volatile Compounds in Goji Juice

The types and contents of volatile compounds (Appendix A) in the fermented goji juice samples and raw juice were found to be different (Figure 1). In total, 38 compounds were identified in the raw juice, whereas 46, 43, and 53 compounds were detected in the goji juice samples fermented with SLV, SZP, and SZVP, respectively. The volatile compounds in the samples mainly belonged to the following classes: acids (5), alcohols (15), ketones (8), aldehydes (13), esters (6), terpenes (3), phenols (5), and others (7).

The acids comprised a class of volatile compounds that increased in abundance after fermentation. Short-chain fatty acids comprising acetic acid, heptanoic acid, and octanoic acid were produced by fermentation with various bacterial strains. In particular, acetic acid was produced via the metabolism of carbohydrates, fats, and amino acids by lactic acid bacteria [17]. The enrichment of this compound contributed to the aroma of fermented juice [18], with a cheesy, fatty, or pungent odor [19]. The contents of these compounds in the goji juice samples fermented with mixtures of strain were in the order of: SZP (48.55 µg/100 g) > SZVP (40.13 µg/100 g) > SLV (32.02 µg/100 g). After fermentation, the concentration of n-hexadecanoic acid increased 5.80–7.47 times compared with the raw juice. However, the threshold for detecting this substance was high, so its contribution to the aroma was not great.

Alcohols, such as 1-octanol, linalool, (*Z*)-3-nonen-1-ol, 1-nonanol, levomenthol, geraniol, and 2,4-decadien-1-ol were produced by different strains during fermentation. In particular, 1-octanol has had an aroma of mushrooms and linalool had a floral odor [20]. Dimethyl-silanediol, 1-hexanol, benzyl alcohol, and phenylethyl alcohol were detected in the fermented goji juice samples and raw juice. The benzyl alcohol and phenylethyl alcohol contents were significantly increased after fermentation, where their concentrations followed the order of: SZVP (76.86 µg/100 g) > SZP (55.87 µg/100 g) > SLV (54.71 µg/100 g). These chemicals have floral odors and the OAVs were greater than 1 [20], indicating their contributions to the overall flavor of the goji juice.

Acetoin and (*E*)-1-(2,6,6-trimethyl-1,3-cyclohexadien-1-yl)-2-buten-1-one were only present in the fermented juice. Acetoin can be converted from 2,3-butanedione by diacetyl reductase, and it can also be produced via citrate metabolism [21], where it had a creamy or vanilla odor [22]. Other ketones were present in the fermented goji juice samples and raw juice, where their contents were increased after bacterial fermentation. The concentrations of 6-methyl-5-hepten-2-one and trans-β-ionone were high in the fermented goji juice samples. It was found that 6-methyl-5-hepten-2-one had a green odor and trans-β-ionone had a violet fragrance [23], and these chemicals contributed to the aroma of the fermented goji juice wine [19]. Thus, the increased contents of these chemicals in the present study may positively affect the aroma of the fermented goji juice. The combined concentrations of 6-methyl-5-hepten-2-one and trans-β-ionone followed the order of: SLV (67.69 µg/100 g) > SZVP (64.66 µg/100 g) > SZP (54.17 µg/100 g).

Nonanal was present in the raw juice but absent after fermentation. The hexanal content was also reduced by fermentation compared with the raw juice, which is consistent with a previous report showing that fermentation reduced the hexanal content in soy milk [24]. These compounds smell fatty and grassy, and they lead to the production of a rancid odor as their concentrations increase [25]. In the present study, the odor was not enhanced in the goji juice after fermentation. The benzaldehyde, benzeneacetaldehyde, (*E*)-2-octenal, (*E*)-2-nonenal, and 3,5-dimethyl-benzaldehyde contents were increased after fermentation. Benzaldehyde provides a cherry aroma as its concentration increases [19]. Among the fermented samples, the (*E*,*E*)-2,6-nonadienal content only increased significantly in the juice fermented with SLV, and it was not detected in the juices fermented with other mixtures. This compound may be metabolized by *L. reuteri* or decomposed or transformed by *L. plantarum* or *B. licheniformis*. 

The concentration of 2-pentyl-furan, a fatty acid oxidation product [26], increased by 8.72, 6.44, and 11.63 times after being fermented with SLV, SZP, and SZVP, respectively. Zhang et al. [27] found that the enzymes produced by *Aspergillus* in fermentation of tea increased the content of 2-pentyl-furan, thereby enhancing the floral and grassy odors to improve the sensory quality. Therefore, it is proposed that the metabolites produced by the fermentation strains may increase the 2-pentyl-furan content, where the metabolites generated by *B. velezensis* may be important. The concentration of 2-methoxy-4-vinylphenol increased by 6.29, 5.42, and 6.90 times after fermentation with SLV, SZP, and SZVP, respectively. The odor threshold is low for 2-methoxy-4-vinylphenol and, thus, it made significant contributions to the aroma of fermented products, such as red wine [28].

Overall, the volatile compounds had positive effects on the aroma, and their contents were higher in the goji juice samples fermented with SZVP and SLV. The higher alcohol and ketone contents in the fermented goji juice may be responsible for the strongly floral and fruity aroma. In addition to the volatile compounds comprising alcohols, acids, aldehydes, and ketones, John et al. [29] found that some amino acids can contribute to the aroma. 

### 2.3. Phenolic Antioxidants and Structure–Activity Relationships in Goji Juice

#### 2.3.1. Total Phenolics and Flavonoid Contents

The contents of total phenolics increased in the goji juice samples after fermentation (*p* < 0.05) (Figure 2a). The concentration of total phenolics was 467.13 ± 1.73 mg/100 mL in the raw juice but increased to 1.27, 1.11, and 1.07 times in the goji juice samples fermented with SLV, SZP, and SZVP, respectively. Previous studies have shown that fermentation generally increased the content of phenolics. For example, Ricci et al. [30] fermented elderberry juice with *L. rhamnosus*, *L. plantarum*, and *L. casei*, and found that the content of total phenolics increased after fermentation. The acid substances produced by *L. plantarum* during fermentation increased the content of total phenolics in broccoli [31]. The finding is consistent with the results obtained in the present study. It is proposed that enzymes and carboxylic acids produced by the fermentation process may have destroyed the structure of the cells to release phenolic substances, thereby increasing the total phenol content after fermentation.

The flavonoid contents varied among the raw juice and the juice samples fermented with SZP and SZVP, and the flavonoid content was the highest in the juice fermented with SLV compared with the other samples (Figure 2b). It was reported that fermentation increased the contents of flavonoids in wheat and corn straw extracts, which may be related to the β-glucosidase produced by *L. plantarum* during fermentation [32]. However, in the present study, the flavonoid contents in the juice samples fermented with SZP and SZVP containing *L. plantarum* differed clearly from that in the raw juice, possibly due to the interactions among metabolites produced during fermentation by each strain.

#### 2.3.2. Extractable and Bound Individual Phenolics

Phenolics are among the most frequently studied bioactive compounds in fermented juices. In the present study, we identified 10 bioactive phenolics comprising five phenolic acids and five flavonoids by high-performance liquid chromatography in the free and bound extracts from the goji juice samples (Table 2). 

All of the goji juice samples contained five types of phenolic acids, comprising *p*-hydroxybenzoic acid, chlorogenic acid, caffeic acid, *p*-coumaric acid, and ferulic acid. In the raw goji juice and fermented goji juice samples, the contents of the free forms of *p*-hydroxybenzoic acid, chlorogenic acid, and *p*-coumaric acid were higher than those of the bound forms. Compared with the raw goji juice, the concentrations of the free forms of *p*-hydroxybenzoic acid and chlorogenic acid were increased by 11.1–372.64% in the fermented juice samples. The bound forms of *p*-hydroxybenzoic acid and chlorogenic acid were not detected in the juices fermented with SLV and SZVP. Fermentation can promote the hydrolysis of chlorogenic acid to yield caffeic acid [5]. The concentration of the free form of *p*-coumaric acid was increased to 1.81 times with SLV, whereas almost none of the free form of *p*-coumaric acid was detected in the other two fermented samples, possibly due to the presence of *L. plantarum* in SZP and SZVP. It was found that phenolic acid decarboxylase metabolized *p*-coumaric acid into phloretic acid or *p*-hydroxybenzoic acid. *Lactobacillus plantarum* contains the *p*-coumarate decarboxylase gene [33]. In addition, the concentrations of the bound forms of caffeic acid and ferulic acid were higher in the raw goji juice compared with the free forms. The concentration of the free form of caffeic acid increased after fermentation with SLV and SZP, possibly due to the hydrolysis of chlorogenic acid. However, the concentration of the bound form of caffeic acid decreased by 1.29 times with SLV, suggesting the further bacterial metabolization of caffeic acid [5]. Compared with the raw goji juice, the concentrations of the free from of ferulic acid increased by 13.23–14.77 times in the fermented juice samples. Fermentation may have promoted the structural breakdown of plant cell walls, thereby leading to the increases in the concentration of free phenolic compounds [34].

All of the goji juice samples contained five types of flavonoids, comprising rutin, quercentin, naringenin, luteolin, and kaempferol. In the raw and fermented goji juice samples, the concentrations of the free forms of rutin and quercentin were higher than the bound forms. Compared with the raw goji juice, the concentrations of the free form of rutin were increased by 1.35, 1.29, and 3.10 times in the juice samples fermented with SLV, SZP, and SZVP, respectively. Similar results were obtained for other fermented juices in a previous study [35]. The increases in the concentrations of the free form of rutin might not have been due to the conversion of the bound form of rutin because the concentration of the bound form of rutin was low in the goji juice. Thus, the increases may have been due to the depolymerization of high molecular weight phenolics by (poly)phenoloxidases during fermentation [35]. The free form of naringenin was not detected in the raw goji juice, but it appeared after fermentation. In addition, the concentration of the bound form of naringenin decreased by 51.72% in the fermented juice, possibly due to the conversion of the bound form of naringenin. The free forms of luteolin and kaempferol were not detected in the raw and fermented goji juice samples. However, the content of the bound form of luteolin increased after fermentation. Kwaw et al. [35] reported that the stability of phenols depended on the pH. The pH was reduced after fermentation (3.84–4.20), and this might have stabilized the bound forms of luteolin and kaempferol [35].

#### 2.3.3. Free Radical Scavenging Activity and Structure-Activity Relationships

The antioxidant capacities of the goji juice samples were determined using two methods based on different principles comprising the 2,2-diphenyl-1-pricrylhydrazyl radical (DPPH) and the hydroxyl radical assay. The free radical scavenging rates in the juice samples ranged from 58.52 ± 3.28% to 80.21 ± 1.17% and 70.48 ± 5.65% to 82.39 ± 5.94% according to the DPPH and hydroxyl radical assays, respectively (Figure 3). 

The DPPH free radical scavenging rate increased with fermentation (Figure 3a). There were no obvious differences in the DPPH free radical scavenging rates among the three fermentation mixtures (*p* > 0.05), suggesting that fermentation promoted the release of antioxidant compounds. In addition, some bacterial strains have antioxidant capacities. For example, Xing et al. [36] found that *L. rhamnosus* had a high DPPH free radical scavenging ability. In the present study, each fermentation mixture contained *L. rhamnosus* and the presence of this strain may have improved the DPPH free radical scavenging activity of the fermented juice. Xu et al. [37] reported that the extracellular polysaccharide produced by *L. plantarum* exhibited an antioxidant activity, where the DPPH free radical scavenging rate of the purified exopolysaccharide was more than 80%.

The hydroxyl radical scavenging rate of the raw goji juice was 82.39% (Figure 3b). There were no major differences in the hydroxyl radical scavenging rates among the fermented and raw juice samples. Zeng et al. [38] also obtained similar results after fermentation using a combination of lactic acid bacteria containing *L. plantarum* and *L. rhamnosus*, which did not increase the hydroxyl radical scavenging rate in yam juice. These strains might not have altered the substances in the goji juice that act on hydroxyl radicals. Thus, fermentation did not enhance the hydroxyl radical scavenging rate of the goji juice, but the fermented juice samples still had high rates.

The fermentation of the goji juice with each mixture clearly increased the DPPH free radical scavenging rate, and high hydroxyl radical scavenging rates were maintained, possibly due to the presence of different phenolic compounds that act on different free radicals. Different antioxidant indexes were correlated with the concentrations of the free and bound forms of phenolic compounds (Table 3). Among all of the phenolic compounds detected in the present study, the free *p*-hydroxybenzoic acid, chlorogenic acid, ferulic acid, rutin, and naringenin had high positive correlations with the DPPH free radical scavenging rate. The concentrations of these free fractions in the fermented goji juices were higher than those in the unfermented goji juice, which may explain why the DPPH free radical scavenging rates were higher in the fermented juices. However, among all of the bound forms of the phenols detected, only that of *p*-coumaric acid was positively correlated with the DPPH free radical scavenging rate. Our results agree with those reported by Shen et al. [39], who demonstrated the major contribution of the bound form of *p*-coumaric acid to the antioxidant capacity. None of the phenolic compounds detected in the present study had high positive correlations with the hydroxyl radical scavenging rate, except for the bound form of chlorogenic acid. The poor correlations between the phenolic substances and hydroxyl radicals may be related to the lack of obvious differences in the hydroxyl radical scavenging rate among the four samples. The antioxidant activities were affected by the molecular structure of phenols [35] and phenolic compounds have different sites that are active with diverse free radicals. Thus, the structures of the phenolic compounds that act on hydroxyl radicals might not have changed during the fermentation process. 

## 3. Materials and Methods 

### 3.1. Materials

Dried goji fruits were purchased from Zhongning, Ningxia, China. Pectinase and baking soda were purchased from a local supermarket (Hangzhou, China). Sucrose was from Taikoo Sugar Co., Ltd. (Shanghai, China). The MRS (De Man, Rogosa, Sharpe) and LB (Luria-Bertani) media were from Qingdao Hope Biotechnology Co., Ltd. (Qingdao, China). The BCA protein concentration assay kit was purchased from Beyotime Biotechnology Co., Ltd. (Shanghai, China). Methanol and formic acid (HPLC grade) were from Sigma–Aldrich (St. Louis, MO, USA). Chlorogenic acid, *p*-hydroxybenzoic acid, chlorogenic acid, caffeic acid, *p*-coumaric acid, ferulic acid, rutin, quercentin, naringenin, luteolin, kaempferol, and cyclohexanone were purchased from Aladdin (Shanghai, China). All other reagents of analytical reagent (AR) grade were purchased from Sinopharm Chemical Reagent Company (Shanghai, China).

### 3.2. Bacterial Strains

Three strains of lactic acid bacteria (*L. plantarum*, *L. rhamnosus*, and *L. reuteri*) and two *Bacillus* species (*B. velezensis* and *B. licheniformis*) were used to ferment goji juice. *Lactobacillus plantarum* and *L. rhamnosus* were preserved in the Ningxia Key Laboratory for Food Microbial-Applications Technology and Safety Control, and the other strains were isolated from naturally fermented foods. Single colonies were placed into 15 mL of MRS or LB medium and incubated for 24 h under anaerobic conditions at 37 °C.

### 3.3. Fermentation of Goji Juice

A mixture of dried fruit and water at a ratio of 1:5 (*w*/*v*) was homogenized after soaking for 12 h. Next, 0.15% pectinase (*w*/*v*) was added to the goji juice and incubated at 50 °C for 2 h. The seeds were removed by filtering through a gauze and 5% sucrose was added, before the pH was adjusted to 6.5 with baking soda. After pasteurizing the goji juice, it was cooled to room temperature. The bacterial mixtures were prepared according to the ratios shown in Table 4 and added to the juice as an inoculum at 0.5% (*v*/*v*), before fermenting for 20 h under anaerobic conditions at 37 °C. Non-fermented goji juice was treated as a control sample.

### 3.4. Analysis of Basic Components

Reducing sugars were determined according to the DNS (3,5-dinitro-2-hydroxybenzoic acid) method [40]. Lactic acid contents were measured with an SBA-40E biosensor analyzer. The protein contents were determined using a BCA protein concentration assay kit. The total phenol [39,41] and flavonoid [39,42] contents were determined according to a previously reported method. The DPPH free radical scavenging and hydroxyl radical scavenging activities were determined as described previously [43].

The number of live bacteria was determined according to the ISO 4833-2003. Briefly, the sample was diluted appropriately, then 1 mL of each diluted solution was added to the sterilized plate count agar (PCA) (3 parallels per dilution), after which they were incubated at 37 °C for 48 h and then the number of colonies was counted. The formula to calculate the number of live bacteria in the sample is as follows: *N* = ∑C/(n_1_ + 0.1 × n_2_) × dwhere *N* means the number of live bacteria, C means the number of colonies in two consecutive dilution plates, n_1_ means the number of low dilution multiple plates, n_2_ means the number of high dilution multiple plates, and d means the dilution factor (the low dilution multiple).

### 3.5. Assessment of Sensory Quality

Sensory quality assessments were conducted by a team comprising six food quality professionals. Sensory evaluations were performed based on the color, texture, and flavor of the fermented samples. Samples were randomly coded with three digits and presented at intervals of 2 min per sample, where the testers rinsed their mouths with water between samples. The scoring criteria are shown in Table 5.

### 3.6. Analysis of Volatile Compounds

A sample weighing 3 ± 0.001 g was placed into a 15 mL glass vial, and 2 µL of cyclohexanone (0.25 µg/g) was added as an internal standard. Headspace micro-extraction was performed with a SPME (50/30 µm divinylbenzene/carboxen/polydimethylsiloxane; DVB/CAR/PDMS) fiber for 30 min at 60 °C after equilibrating for 15 min. After extraction, the analytes were desorbed in a gas chromatography mass spectrometry injector at 240 °C for 3 min. Samples were analyzed using helium (99.99%) as the carrier gas at a flow rate of 1.2 mL/min in the splitless injection mode. The programmed temperature was held at 50 °C for 2 min and then increased at a rate of 5 °C/min to 180 °C and maintained for 10 min, and then increased at a rate of 10 °C/min to 260 °C. The volatile compounds were identified with a mass-spectral library (NIST 14) and matched against the references. 

### 3.7. Extraction and Separation of Free Phenolics

Phenolic compounds were extracted as described by Shen et al. [39]. Briefly, 10 mL of each sample was extracted with 30 mL ethanol (80%) for 15 min at 4 °C, followed by centrifugation at 2500× *g* (centrifugal force) for 10 min. The residue was extracted again twice. The supernatants were combined, evaporated to dryness at 45 °C, and reconstituted to 10 mL with methanol. The free phenolics were analyzed in each sample. The residues obtained after extracting the free phenolics were digested in 30 mL of 2 mol/L NaOH for 1 h at 25 °C with shaking. The samples were then neutralized with concentrated hydrochloric acid and extracted three times with ethyl acetate. The ethyl acetate extracts were combined, evaporated to dryness at 45 °C, and reconstituted to 10 mL of methanol. The contents of bound phenolics were determined in each sample. Each sample was tested in triplicate.

The phenolic acid and flavonoid contents were determined with a Waters system as described by Shen et al. [39] with some modifications. Briefly, samples were separated by chromatography with a ZORBAX SB-C18 column (250 mm × 4.6 mm, i.d., 5 µm). The flow rate of the mobile phase was 1 mL/min. The column temperature was 30 °C and the injection volume was 20 µL. The detection wavelengths were 288 nm and 360 nm. Mobile phase A was methanol and mobile phase B was 0.1% formic acid *(v*/*v*). The solvent gradient elution program was as follows: 15–20% A (0–8 min), 20–60% A (8–18 min), 60–80% A (18–30 min), 80% A (30–40 min), and 80–15% A (40–50 min). The phenolic acid and flavonoid contents were calculated based on linear calibration curves obtained with standards and expressed as mg/100 mL goji juice.

### 3.8. Statistical Analysis

Data were analyzed applying the Statistical Program for Social Sciences (SPSS 20.0, Chicago, IL, USA). Significant differences among the samples were discriminated using one-way ANOVA, applying the LSD post-hoc test. A *p*-value of less than 0.05 was considered to indicate a significant difference. Data were expressed as the means standard deviation. Data were plotted graphically used OriginPro (version 8.5; OriginLab Corporation, Northampton, MA, USA) and R software (version 3.5.1; The R Foundation for Statistical Computing, Vienna, Austria).

## 4. Conclusions

In the present study, SLV (*Lactobacillus rhamnosus*, *Lactobacillus reuteri*, and *Bacillus velezensis*), SZP (*Lactobacillus rhamnosus*, *Lactobacillus plantarum*, and *Bacillus licheniformis*) and SZVP (*Lactobacillus rhamnosus*, *Lactobacillus plantarum*, *Bacillus velezensis*, and *Bacillus licheniformis*) were used to ferment goji juice. After fermentation, the compositions of the goji juices had obvious differences, including the concentrations of reducing sugars, lactic acid, proteins, volatile substances, and phenols. Both SLV and SZVP had higher fermentation capacities and greater positive effects on the aroma. In addition, all of the fermentation mixtures enhanced the phenol concentrations in the juice, and the antioxidant capacities were compared using DPPH and hydroxyl radical scavenging assays. Fermentation modified the phenolic composition of the goji juice, and the antioxidant capacity was strongly correlated with the content of the free forms of phenols. According to our results, SLV and SZVP can possibly be used as starters in goji juice fermentation. 

## Figures and Tables

**Figure 1 molecules-24-03519-f001:**
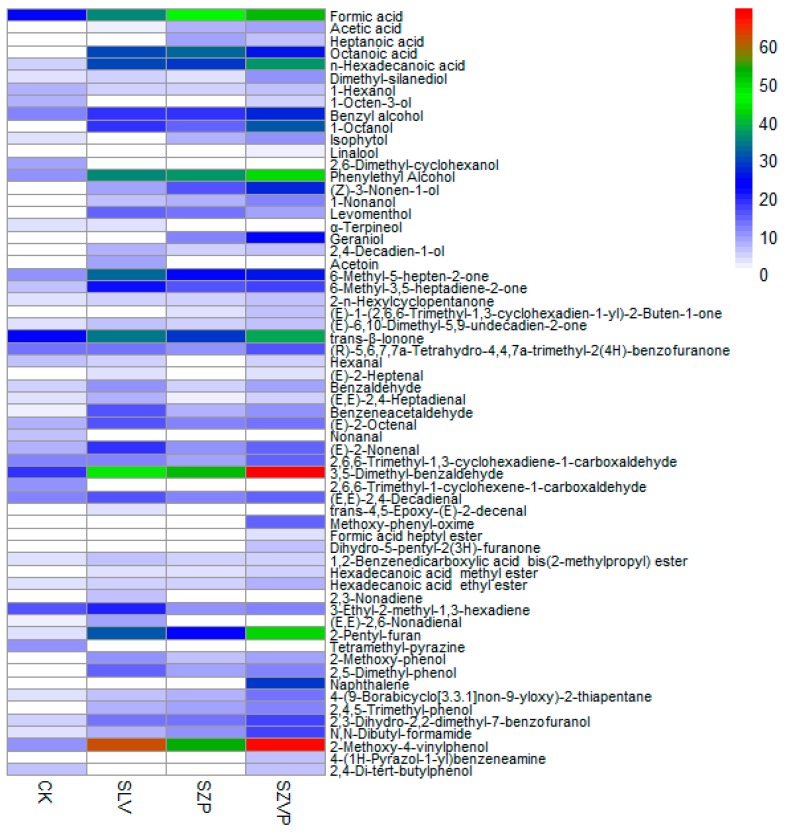
The contents of volatile compounds in the fermented goji juice by different bacteria (µg/100 g). The heat map was based on the contents of volatile compounds detected in fermented and unfermented goji juice. The scale ranging from 0 (white) to 70 (red) was used to show the contents of volatile matters (µg) in 100 g goji juice.

**Figure 2 molecules-24-03519-f002:**
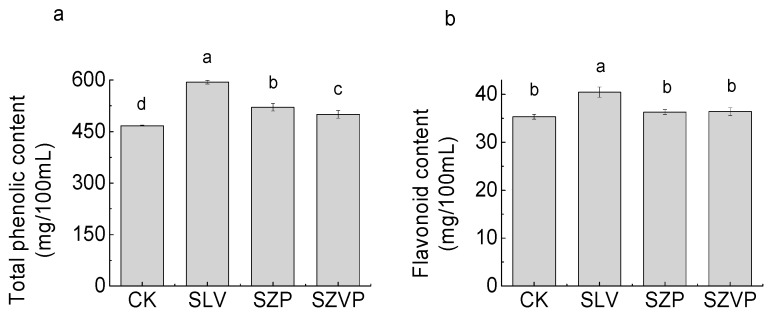
Contents of total phenols (**a**) and flavonoids (**b**) in raw juice and fermented goji juices. Different superscript letters (a–d) in the columns indicate significant difference (*p* ≤ 0.05).

**Figure 3 molecules-24-03519-f003:**
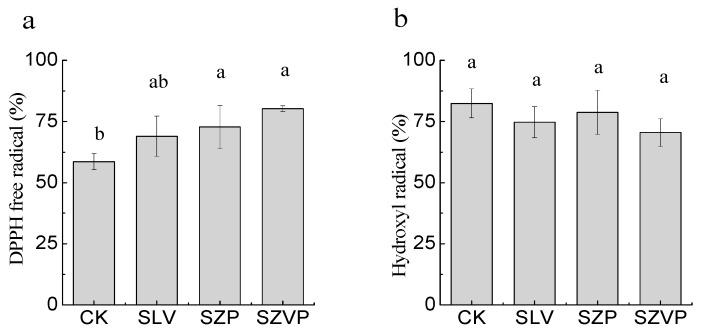
The DPPH free radical scavenging rate (**a**) and hydroxyl radical scavenging rate (**b**) of raw juice and fermented goji juices. Different superscript letters (a,b) in every column indicate significant difference (*p* ≤ 0.05).

**Table 1 molecules-24-03519-t001:** Comparison of fermentation characteristics of each composite strain group.

**Samples**	**Reducing Sugars** **(mg/100 mL)**		**Lactic Acid** **(mg/100 mL)**		**Protein** **(mg/100 mL)**		**Number of Live Bacteria (CFU/mL)**	
CK	5521.04 ± 4.58 ^a^		25.30 ± 0.71 ^d^		39.23 ± 0.67 ^d^		-	
SLV	4884.23 ± 9.04 ^b^		316.93 ± 7.31 ^b^		74.75 ± 1.86 ^b^		8.43 × 10^9^	
SZP	4603.91 ± 9.35 ^c^		233.80 ± 10.97 ^c^		51.46 ± 0.35 ^c^		2.17 × 10^10^	
SZVP	3951.46 ± 13.73 ^d^		422.14 ± 6.52 ^a^		82.74 ± 0.31 ^a^		2.23 × 10^10^	
**Samples**	**Sensory Quality**							
	**Color**	**Flavor**		**Taste**		**Acceptability**		**Total**
CK	23.50 ± 1.38 ^a^	22.83 ± 2.04 ^a^		13.83 ± 1.72 ^c^		13.20 ± 1.64 ^b^		73.17 ± 5.04 ^b^
SLV	23.50 ± 1.05 ^a^	21.50 ± 1.97 ^a^		23.17 ± 2.32 ^a^		16.20 ± 2.68 ^a^		84.17 ± 2.32 ^a^
SZP	23.00 ± 0.63 ^a^	20.67 ± 2.58 ^a^		20.33 ± 2.73 ^b^		13.80 ± 1.64 ^b^		77.50 ± 4.18 ^b^
SZVP	22.83 ± 0.98 ^a^	20.83 ± 2.32 ^a^		21.17 ± 1.72 ^ab^		15.20 ± 1.79 ^ab^		80.00 ± 2.97 ^ab^

CK means unfermented goji juice. SLV means goji juice fermented using *L. rhamnosus*, *L. reuteri*, and *B. velezensis*. SZP means goji juice fermented using *L. rhamnosus*, *L. plantarum*, and *B. licheniformis*. SZVP means goji juice fermented using *L. rhamnosus*, *L. plantarum*, *B. velezensis*, and *B. licheniformis*. The data were expressed as the mean value ± standard deviation, except for the number of live bacteria. Data with different superscript letters (a–d) in the same column were significantly different (*p* ≤ 0.05) and ‘-’ means data not detected.

**Table 2 molecules-24-03519-t002:** The contents of extractable and bound phenolics in goji juices (mg/100 mL).

Phenolics		CK	SLV	SZP	SZVP
**Phenolic Acids**					
	Free	4.24 ± 2.24 ^c^	9.09 ± 0.52 ^b^	6.44 ± 2.97 ^b^	20.04 ± 0.54 ^a^
*p*-Hydroxybenzoic acid	Bound	1.81 ± 0.55 ^a^	- ^b^	- ^b^	- ^b^
	Total	6.05 ± 2.79 ^b^	9.09 ± 0.52 ^b^	6.44 ± 2.97 ^b^	20.04 ± 0.54 ^a^
	Free	0.99 ± 0.02 ^b^	1.10 ± 0.06 ^b^	1.32 ± 0.67 ^b^	2.41 ± 0.10 ^a^
Chlorogenic acid	Bound	0.18 ± 0.04 ^b^	- ^c^	0.21 ± 0.00 ^a^	- ^c^
	Total	1.17 ± 0.06 ^b^	1.10 ± 0.06 ^b^	1.53 ± 0.67 ^b^	2.41 ± 0.10 ^a^
	Free	0.62 ± 0.23 ^a^	0.95 ± 0.27 ^a^	0.91 ± 0.32 ^a^	0.48 ± 0.01 ^b^
Caffeic acid	Bound	0.72 ± 0.24 ^b^	0.51 ± 0.01 ^c^	0.98 ± 0.02 ^a^	0.94 ± 0.02 ^a^
	Total	1.33 ± 0.41 ^b^	1.47 ± 0.26 ^ab^	1.89 ± 0.30 ^a^	1.42 ± 0.02 ^b^
	Free	2.42 ± 0.03 ^b^	4.38 ± 0.01 ^a^	0.07 ± 0.02 ^c^	- ^d^
*p*-Coumaric acid	Bound	0.03 ± 0.01 ^c^	- ^d^	0.52 ± 0.01 ^a^	0.40 ± 0.01 ^b^
	Total	2.45 ± 0.03 ^b^	4.38 ± 0.01 ^a^	0.59 ± 0.02 ^c^	0.40 ± 0.01 ^d^
	Free	0.13 ± 0.14 ^c^	1.73 ± 0.05 ^b^	1.92 ± 0.14 ^a^	1.72 ± 0.01 ^b^
Ferulic acid	Bound	0.54 ± 0.02 ^b^	0.42 ± 0.03 ^c^	1.21 ± 0.03 ^a^	0.33 ± 0.03 ^d^
	Total	0.67 ± 0.12 ^c^	2.15 ± 0.06 ^b^	3.12 ± 0.15 ^a^	2.05 ± 0.03 ^b^
**Flavonoids**					
	Free	4.88 ± 0.94 ^c^	6.59 ± 0.26 ^b^	6.29 ± 0.93 ^b^	15.13 ± 0.60 ^a^
Rutin	Bound	0.08 ± 0.05 ^b^	- ^c^	0.21 ± 0.11 ^a^	0.05 ± 0.02 ^b^
	Total	4.96 ± 0.95 ^c^	6.59 ± 0.26 ^b^	6.49 ± 1.01 ^b^	15.17 ± 0.62 ^a^
	Free	0.99 ± 0.10 ^a^	0.27 ± 0.04 ^d^	0.48 ± 0.02 ^c^	0.86 ± 0.06 ^b^
Quercentin	Bound	0.37 ± 0.01 ^b^	0.11 ± 0.03 ^c^	0.42 ± 0.09 ^a^	0.42 ± 0.02 ^a^
	Total	1.36 ± 0.11 ^a^	0.38 ± 0.04 ^c^	0.90 ± 0.09 ^b^	1.28 ± 0.07 ^a^
	Free	- ^d^	1.34 ± 0.08 ^c^	6.23 ± 0.07 ^a^	4.49 ± 0.24 ^b^
Naringenin	Bound	0.58 ± 0.06 ^a^	0.28 ± 0.03 ^b^	- ^c^	- ^c^
	Total	0.58 ± 0.06 ^d^	1.63 ± 0.11 ^c^	6.23 ± 0.07 ^a^	4.49 ± 0.24 ^b^
	Free	- ^a^	- ^a^	- ^a^	- ^a^
Luteolin	Bound	0.61 ± 0.02 ^d^	0.72 ± 0.04 ^b^	1.32 ± 0.04 ^a^	0.67 ± 0.01 ^c^
	Total	0.61 ± 0.02 ^d^	0.72 ± 0.04 ^b^	1.32 ± 0.04 ^a^	0.67 ± 0.01 ^c^
	Free	- ^a^	- ^a^	- ^a^	- ^a^
Kaempferol	Bound	0.30 ± 0.03 ^a^	0.33 ± 0.03 ^a^	0.26 ± 0.02 ^b^	0.24 ± 0.06 ^b^
	Total	0.30 ± 0.03 ^a^	0.33 ± 0.03 ^a^	0.26 ± 0.02 ^b^	0.24 ± 0.06 ^b^

The data are expressed as the mean value ± standard deviation. The data with different superscript letters (a–d) in the same row indicate significant difference (*p* ≤ 0.05), and ‘-’ means not detected.

**Table 3 molecules-24-03519-t003:** Pearson’s correlation coefficients of phenolic compounds and antioxidant capacities.

	***p*-Hydroxybenzoic Acid**		**Chlorogenic Acid**		**Caffeic Acid**		***p*-Coumaric Acid**		**Ferulic Acid**	
	Free	Bound	Free	Bound	Free	Bound	Free	Bound	Free	Bound
DPPH	0.639 **	−0.783 **	0.711 **	−0.411	−0.091	0.400	−0.503 *	0.619 *	0.700 **	−0.031
HFR	−0.548 *	0.490	−0.565 *	0.541 *	0.039	0.062	0.128	−0.188	−0.459	0.342
	**Rutin**		**Quercentin**		**Naringenin**		**Luteolin**		**Kaempferol**	
	Free	Bound	Free	Bound	Free	Bound	Free	Bound	Free	Bound
DPPH	0.728 **	0.288	−0.187	0.142	0.672 **	−0.803 **	0.000	0.246	0.000	−0.363
HFR	−0.615 *	0.057	0.110	0.236	-0.241	0.454	0.000	0.098	0.000	0.254

DPPH means 2,2-diphenyl-1-pricrylhydrazyl free radical scavenging power; HFR means hydroxyl radical scavenging power; Data with ‘**’ and ‘*’ means statistically significant at *p* ≤ 0.01 and *p* ≤ 0.05, respectively.

**Table 4 molecules-24-03519-t004:** Different combinations of microbial strains used in the study.

Sample Name	Strain Combination
SLV	*L. rhamnosus*, *L. reuteri*, *B. velezensis*
SZP	*L. rhamnosus*, *L. plantarum*, *B. licheniformis*
SZVP	*L. rhamnosus*, *L. plantarum*, *B. velezensis*, *B. licheniformis*

Every bacteria were mixed at the same ratio.

**Table 5 molecules-24-03519-t005:** Standard of sensory quality for fermented goji juice.

Sensory Quality	Score	Standard
Color	19–27	The color is moderate, very good.
	10–18	Color is slightly darker or lighter, acceptable.
	1–9	The color is too deep or light, very bad.
Flavor	19–27	The aroma of the goji juice is pure and has no odor.
	10–18	The aroma of the goji juice is obvious, slightly odor.
	1–9	The aroma of the goji juice is light and the odor is heavy.
Taste	19–27	Acidity and sweetness are suitable.
	10–18	Slightly acidic or sweet.
	1–9	Too acidic or too sweet.
Acceptability	0	Extremely dislikes.
	3	Very dislike.
	6	Slightly dislikes.
	9	Indifferent.
	12	Slightly like.
	15	Very like.
	18	Extremely like.

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
