# Peer review of "Fermentation by Multiple Bacterial Strains Improves the Production of Bioactive Compounds and Antioxidant Activity of Goji Juice"

_molecules, 2019, doi:10.3390/molecules24193519_

Round 1

Reviewer 1 Report

Manuscript number: molecules-597168

Title: Fermentation by multiple bacterial strains improves the production of bioactive compounds and antioxidant activity of goji juice

The manuscript is well written and scientifically sounds. The aims are well described as well as the methods and results. I suggest few changes and in order to better report the increasing/decreasing data and some cosmetics.

-Increasing/decreasing times: in all the text increasing/decreasing contents were reported as n times of the control sample's content. The present form is a little difficult to be understandable and could induce misunderstandings. For example, at line 77 it was reported that soluble proteins of CK was 0.91 times higher than SLV, that could be understandable in two ways as:

39.23+39.23*0.91

39.23*0.91

in the first case the n of times is representative of the true data, in the second case it actually represents a decrease (39.23*0.91 is < than 39.23). In my point of view, it is better to report 1.91 times (39.23*1.91=74.75) or in percentage (an increasing of 90.54%).

At the same line it is reported that CK soluble proteins are 0.11 times higher than SZVP, that is a mistake in any way as 82.74/39.23=2.11 so that, at least in the present annotation, must be 1.11 times. Anyway, I suggest 2.11 times or in percentage.

The same suggestion is valid for all the text ad also for the decreasing data.

Lines: 26, 77, 180, 210, 213, 221, 222, 229, 230, 237

-Materials and methods

Please report how "number of live bacteria" were determined and on how many samples (in the present form in Table 1 there is no standard deviation for these data, was that performed in single? Why?)

Statistical analysis, please state that you performed an one-way ANOVA and the post-hoc that you used.

-Other needed changes and cosmetics

L178 change > with < as the p value was significative

L182 remove the space before the dot

L308 Bacillus in italic

L316 add a space between 50 and °C

L374 add the bacterial name in brackets within the abbreviations (as made in the abstract).

Reviewer 2 Report

The present manuscript displays a quite interesting topic, but could not be published in its current form, because there exist some major problems in this paper, which I will list as following.

Table 1 - in the case of sensory evaluation, in my opinion in the presentation of the results the individual, studied quality indicators like color, flavor etc should be presented instead of only general "sensory quality".

Figure 1 - this Figure is not clear - it would be better to present these results in the form of a table.

Line 319 “The bacterial mixtures were prepared according to the ratios shown in Table 4” - Table 4 does not specify the ratio at which these bacteria were mixed, but only which bacteria are present in these mixtures - please complete this.

Line 327 “The total phenol and flavonoid contents were determined according to a previously reported methods [39]” - the literature cited is not source literature for the determination of phenolic compounds - please provide original source literature.

The above remark will refer to the whole methodology - please check whether the source literature was cited also in the other methodology points.

Table 2 and all text – rutinum -please check if this name is correct, because in my opinion it should be - rutin

Reviewer 3 Report

Brief summary:

Nowadays, the idea of finding new sources of bioactive compound and the development of functional food and nutraceuticals products represents research directions with great potential and opportunities.

The aim of this study was thus to compare the differences of the goji juices fermented by three mixed strains groups. The properties of fermented goji juices including chemical composition, volatile composition and phenolic compounds were evaluated and compared with a blank sample.

This study addresses an interesting topic and the obtained results provide novel insights into the production of fermented goji juice and the effects of different bacterial strains on some bioactive compounds.

The article is generally well written, the English language is appropriate and understandable, only minor spell check being required. Also, the methods and discussions sections are described very clear, with sufficient details to allow another researcher to reproduce the results.

Prior to publication some minor corrections are required.

Broad comments:

What is the reason for selecting these three strains of bacteria? Could this also produce undesirable effects in the composition of the juice?

Could the process be scaled up at industrial level? Is the process economically feasible? 

Specific comments:

Line 29: I suggest to replace “affected” with another word (i.e. influenced)

Line 36: replace “health food” with “healthy food”

Line 51: I suggest to replace “Little has been reported in the use of strains” with “Not many results…..

Line 351: 2500 g ? I think is rpm

Line 375: The expression “changed greatly” is not recommended to be used in a scientific language.

Round 2

Reviewer 2 Report

No comments